# The Origins and Prevalence of Texture Bias in Convolutional Neural Networks

**Katherine L. Hermann**
Stanford University
hermannk@stanford.edu

**Ting Chen**
Google Research, Toronto
iamtingchen@google.com

**Simon Kornblith**
Google Research, Toronto
skornblith@google.com

## Abstract

Recent work has indicated that, unlike humans, ImageNet-trained CNNs tend to classify images by texture rather than by shape. How pervasive is this bias, and where does it come from? We find that, when trained on datasets of images with conflicting shape and texture, CNNs learn to classify by shape at least as easily as by texture. What factors, then, produce the texture bias in CNNs trained on ImageNet? Different unsupervised training objectives and different architectures have small but significant and largely independent effects on the level of texture bias. However, all objectives and architectures still lead to models that make texture-based classification decisions a majority of the time, even if shape information is decodable from their hidden representations. The effect of data augmentation is much larger. By taking less aggressive random crops at training time and applying simple, naturalistic augmentation (color distortion, noise, and blur), we train models that classify ambiguous images by shape a majority of the time, and outperform baselines on out-of-distribution test sets. Our results indicate that apparent differences in the way humans and ImageNet-trained CNNs process images may arise not primarily from differences in their internal workings, but from differences in the data that they see.

## 1   Introduction

Convolutional neural networks (CNNs) define state-of the-art-performance in many computer vision tasks, such as image classification (57), object detection (83; 40), and segmentation (40). Although their performance in several of these tasks approaches that of humans, recent findings show that CNNs differ in intriguing ways from human vision, indicating fundamental deficiencies in our understanding of these models (87; 70; 32; 23; 76; 37; 3; 44; 46; 1; 48). This paper focuses on one such result, namely that CNNs appear to make classifications based on superficial textural features (36; 4) rather than on the shape information preferentially used by humans (61; 60). Building on a long tradition of work in psychology and neuroscience documenting humans' shape-based object classification, Geirhos et al. (36) compared humans to ImageNet-trained CNNs on a dataset of images with conflicting shape and texture information (e.g. an elephant-textured knife) and found that models tended to classify according to texture (e.g. "elephant"), and humans according to shape (e.g. "knife"). Following their terminology, we call a learner that prefers texture over shape *texture-biased* and one the prefers shape over texture *shape-biased*.

From the point of view of computer vision, texture bias is an important phenomenon for several reasons. First, it may be related to the vulnerability of CNNs to adversarial examples (87), which may exploit features that carry information about the class label but are undetectable to the human visual system (48). Second, a CNN preference for texture could indicate an inductive bias different than that of humans, making it difficult for models to learn human-relevant vision tasks in small-data regimes, and to generalize to different distributions than the distribution on which the model is trained (35).

In addition to these engineering considerations, texture bias raises important scientific questions. ImageNet-trained CNNs have emerged as the model of choice in neuroscience for modeling electrophysiological and neuroimaging data from primate visual cortex (101; 52; 15; 71; 7). Evidence that CNNs are preferentially driven by texture indicates a significant divergence from primate visual processing, in which shape bias is well documented (61; 60; 38). This mismatch raises an important puzzle for human-machine comparison studies. As we show in Section 7, even models specifically designed to match neural data exhibit a strong texture bias.

This paper explores the origins of texture bias in ImageNet-trained CNNs, looking at the effects of data augmentation, training procedure, model architecture, and task. In Section 4, we show that CNNs learn to classify the shape of ambiguous images at least as easily as they learn to classify the texture. So what makes ImageNet-trained CNNs classify images by texture when humans do not? While we find effects of all of the factors investigated, the most important factor is the data itself. Our contributions are as follows:

- We find that naturalistic data augmentation involving color distortion, noise, and blur substantially decreases texture bias, whereas random-crop augmentation increases texture bias. Combining these observations, we train models that classify ambiguous images by shape a majority of the time, without using the non-naturalistic style transfer augmentation of Geirhos et al. (36). These models also outperform baselines on out-of-distribution test sets that exemplify different notions of shape (ImageNet-Sketch (96) and Stylized ImageNet (36)).

- We investigate the texture bias of networks trained with different self-supervised learning objectives. While some objectives decrease texture bias relative to a supervised baseline, others increase it.

- We show that architectures that perform better on ImageNet generally exhibit lower texture bias, but neither architectures designed to match the human visual system nor models that replace convolution with self-attention have texture biases substantially different from ordinary CNNs.

- We separate the extent to which shape information is represented in an ImageNet-trained model from how much it contributes to the model's classification decisions. We find that it is possible to extract more shape information from a CNN's later layers than is reflected in the model's classifications, and study how this information loss occurs as data flows through a network.

## 2   Related work

**Adversarial examples.** Adversarial examples are small perturbations that cause inputs to be misclassified (87; 10; 70). Adversarial perturbations are not entirely misaligned with human perception (29; 106), and reflect true features of the training distribution (48). State-of-the-art defenses on ImageNet use adversarial training (87; 41; 66; 100) or randomized smoothing (17), and images generated by optimizing class confidence under these models are more perceptually recognizable to humans (80; 30; 93; 51). Adversarial training can improve accuracy on out-of-distribution data (99). However, models that are robust to adversarial examples generated with respect to the $\ell_p$ norm are not necessarily robust to other forms of imperceptible perturbations (98; 92).

**Data augmentation and robustness.** Data augmentation is widely used to improve generalization of CNNs on data drawn from the same distribution as the training data (62; 16; 8; 57; 22; 19; 64; 20; 103; 45). Recent work has shown that data augmentation can also improve out-of-distribution robustness (33; 102; 65; 45; 78). However, most of these studies investigate robustness to simple image corruptions, which may not be correlated with robustness to natural distribution shift (88). As with adversarial training (93; 73), augmentation strategies that improve robustness can degrade clean accuracy (65), and tradeoffs exist between robustness to different corruptions (102). Hosseini et al. (47) found that CNNs can learn to generalize to negative (reversed-brightness) images for a held-out class if the training data is augmented with negative images for the other classes.

**Sensitivity of CNNs to non-shape features.** Our work builds on recent evidence for bias for texture versus shape in neural networks based on classification behavior with ambiguous stimuli (36; 4). Other studies have shown that CNNs are sensitive to a wide range of other image manipulations that have little effect on human judgments (32; 5; 23; 76; 37; 3; 44; 46; 1; 6). Moreover, CNNs are relatively *insensitive* to manipulations such as grid scrambling that make images nearly unrecognizable to humans (11), and are far superior to humans at classifying ImageNet images where the foreground object has been removed (107).

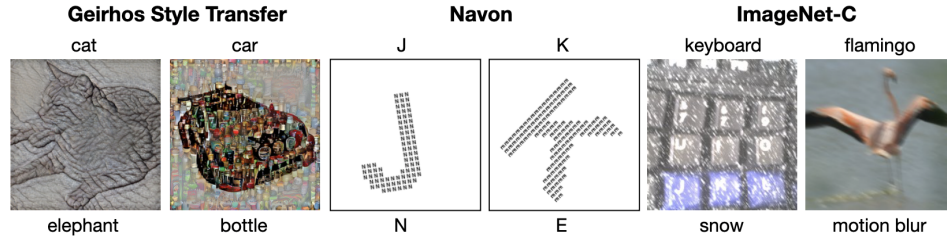

Figure 1: Example items from the three datasets labeled according to shape (top) and texture (bottom). GST items reproduced with permission of (36; 91). ImageNet-C items from (44).

**Similarity of human and CNN perceptual biases.** Despite these differences, ImageNet-trained CNNs appear to share some perceptual biases and representational characteristics with humans. Previous studies found preferences for shape over color (77) and perceptual shape over physical shape (58). Euclidean distance in the representation space of CNN hidden layers correlates well with human perceptual similarity (50; 105), although, when used to *generate* perceptual distortions, simple biologically inspired models are a better match to human perception (9). CNN representations also provide an effective basis for modeling the activity of primate visual cortex (101; 52; 12), even though CNNs' image-level confusions differ from those of humans (74).

# 3 Methods

**Datasets**. In addition to using ImageNet ILSVRC-2012 (79), our experiments used datasets consisting of colored $224 \times 224$ images with independent sets of shape and texture labels. Instead of adopting a single parametric model (e.g. (43; 72; 28; 34)), we construe texture broadly to include natural textures (e.g. dog fur), small units repeating across space, and surface-level noise patterns. Each dataset (Figure 1) captures a possible interpretation of texture. See Appendix E.1.1 for additional discussion of the datasets and their limitations, as well as Figure C.1.

*Geirhos Style-Transfer (GST) dataset*. Introduced by (36), this dataset contains images generated using neural style transfer (34), which combines the content (shape) of one target natural image with the style (texture) of another. The dataset consists of 1200 images rendered from 16 shape classes, with 10 exemplars each, and 16 texture classes, with 3 exemplars each (91).

*Navon dataset*. Introduced by psychologist David Navon in the 1970s to study how people process global versus local visual information (69), Navon figures consist of a large letter ("shape") rendered in small copies of another letter ("texture"). Unlike the GST stimuli, the primitives for shape and texture are identical apart from scale, allowing for a more direct comparison of the two feature types. We rendered each possible shape-texture combination ($26 \times 26$ letters) at 5 positions, yielding a total of 3250 items after excluding items with matching shape and texture. Each image was rotated with an angle drawn from [-45, 45] degrees. In his experiments, Navon found that humans process the shape of these stimuli more rapidly than texture.

*ImageNet-C dataset*. ImageNet-C (44) consists of ImageNet images corrupted by different kinds of noise (e.g. shot noise, fog, motion blur). Here, we take noise type as "texture" and ImageNet class as "shape". The original dataset contains 19 textures each at 5 levels, with 1000 ImageNet classes per level and 50 exemplars per class. To balance shape and texture, for each of 5 subsets of the data (dataset "versions"), we subsampled 19 shapes, yielding a total of 90,250 items per version.

**Shape bias evaluation**. Following (36) we define the *shape bias* of a model as the percentage of the time it classified images from the GST dataset according to shape, provided it classified either shape or texture correctly (see Appendix E.2 for additional details). We call a model *shape-biased* if its shape bias is $> 50\%$, and *texture-biased* if it is $< 50\%$. Throughout, *shape match* and *texture match* indicate the percentage of the time the model chose the image's correct shape or texture label, respectively, as opposed to choosing some other label.

# 4 CNNs can learn shape as easily as texture

Do the texture-driven classifications documented by (36) come from the model, as an inherent property of the CNN architecture or learning process, or from the data, as an accidentally useful

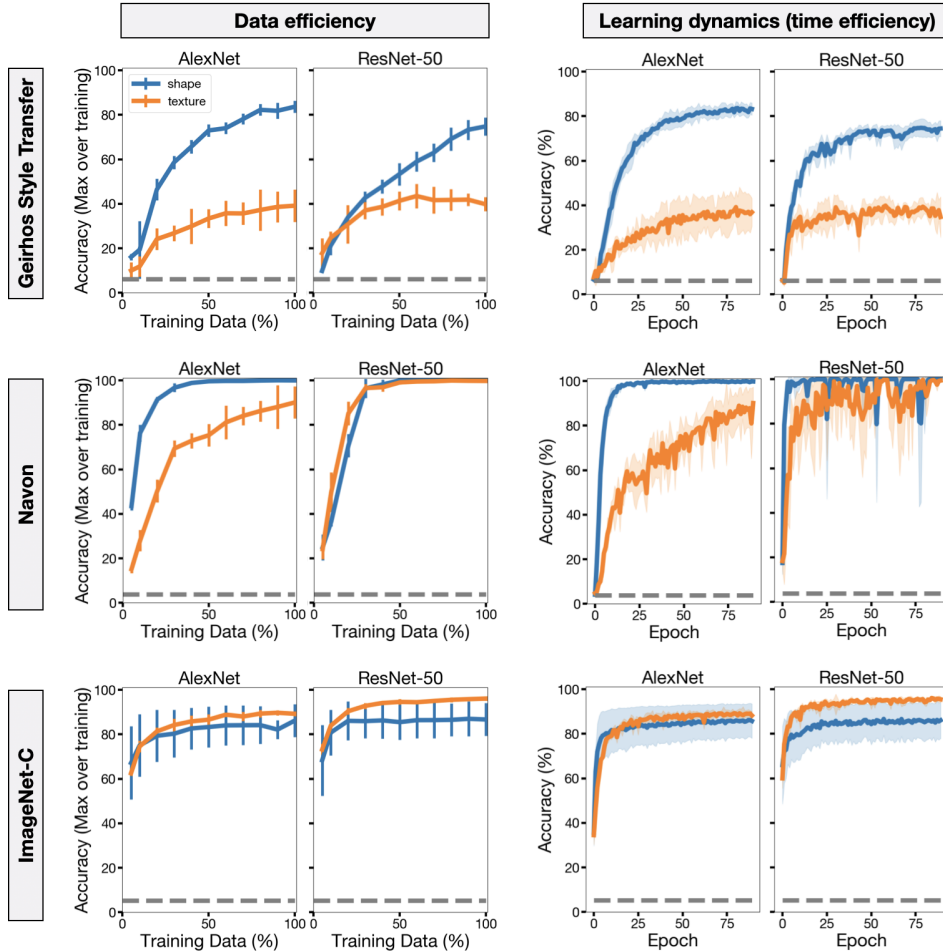

Figure 2: **Task performance of AlexNet and ResNet-50 models as a function of data (first column) and time efficiency (second column), for each dataset (rows). (Data efficiency.)** Performance is the maximum classification accuracy for shape (blue) or texture (orange) over training. All plots show the mean $\pm$ SD across 5 splits. Dashed line indicates chance performance. **(Learning dynamics).** Accuracy over training time on 100% of the training data (GST: 716 items, Navon: 2875 items, ImageNet-C: 80750 items).

regularity that CNNs can exploit? We compared the learning dynamics of models trained to classify ambiguous non-ImageNet images according to either shape or texture: examining the number of training epochs and amount of training data required for models to perform well on these tasks is a way to determine whether shape or texture information is more easily exploited by CNNs. For each of the Geirhos-Style Transfer, Navon, and ImageNet-C datasets, we trained AlexNet (57) and ResNet-50 (42) models to classify images according to (a) shape, or (b) texture, using [5%, 10%, 20%, 30%, ..., 90%, 100%] of the training data. We compared validation accuracies for the two tasks.[1] For details, see Appendix E.1.3.

As shown in Figure 2, nothing prevents AlexNet and ResNet-50 from learning to classify ambiguous images by their shape. We should be cautious in comparing shape and texture performance on GST and ImageNet-C given the unknown in-principle recoverability of shape and texture from these datasets. However, best-possible error rates are known and equal for the synthetic Navon tasks, and we note that in that setting, both models learn to classify by shape at least as accurately as by texture, and, in the case of AlexNet, do so with greater efficiency in time and data. Overall, then, the source of

Table 1: **Random-crop augmentation biases models towards texture.** Characteristics of ImageNet-trained models with random-crop (Random) versus with center-crop (Center) preprocessing. For each model and metric, the preprocessing that achieved a higher value is boldfaced.

| Model | Shape Bias | | Shape Match | | Texture Match | | ImageNet Top-1 Acc. | |
|---|---|---|---|---|---|---|---|---|
| | Random | Center | Random | Center | Random | Center | Random | Center |
| AlexNet | 28.2% | **37.5%** | 16.4% | **19.3%** | **41.8%** | 32.1% | **56.4%** | 50.7% |
| VGG16 | 11.2% | **15.8%** | 7.6% | **10.7%** | **60.1%** | 57.1% | **71.8%** | 62.5% |
| ResNet-50 | 19.5% | **28.4%** | 11.7% | **16.3%** | **48.4%** | 41.1% | **76.6%** | 70.7% |
| Inception-ResNet v2 | 23.1% | **27.9%** | 15.1% | **19.8%** | 50.2% | **51.2%** | **80.3%** | 77.3% |

Table 2: **Color distortion, Gaussian blur, Gaussian noise, and Sobel filtering reduce texture bias.** ResNet-50 models were trained on ImageNet with random crops for 90 epochs. Augmentations were applied with 50% probability. Bolding indicates significantly greater shape bias than the baseline model ($p < 0.05$, permutation test).

| Augmentation | Shape Bias | Shape Match | Texture Match | ImageNet Top-1 Acc. |
|---|---|---|---|---|
| Baseline | 19.5% | 11.7% | 48.4% | 76.6% |
| Rotate 90°, 180°, 270° | 19.4% | 10.8% | 45.1% | 75.7% |
| Cutout | **21.4%** | 12.3% | 45.2% | 76.9% |
| Sobel filtering | **24.8%** | 12.8% | 38.9% | 71.2% |
| Gaussian blur | **25.2%** | 14.1% | 41.7% | 75.8% |
| Color distort. | **25.8%** | 15.3% | 44.2% | 76.9% |
| Gaussian noise | **30.7%** | 17.2% | 38.8% | 75.6% |

texture bias seems to lie in training on ImageNet rather than on CNNs' inductive biases; the remainder of the paper attempts to determine which aspects of ImageNet training are most responsible.

# 5 The role of data augmentation in texture bias

**Random-crop data augmentation increases texture bias**. In their experiments showing a texture bias in ImageNet-trained CNNs, Geirhos et al. (36) followed the standard practice of doing random-crop augmentation: crop shapes are sampled as random proportions of the original image size from [0.08, 1.0] with aspect ratio sampled from [0.75, 1.33], and then resized to $224 \times 224$px (90). We hypothesized that random-crop augmentation might remove global shape information from the image, since for large central objects, randomly varying parts of the object's shape may appear in the crop, rendering shape a less reliable feature relative to texture. We tested whether this was the case by comparing shape bias in random-crop and center-crop versions of AlexNet (57), VGG16 (86), ResNet-50 (42; 89; 49), and Inception-ResNet v2 (84). We evaluated shape bias using the GST images as described in Section 3 (see Appendix E.3 for details and additional discussion).

Center-crop augmentation reduced texture bias relative to random-crop augmentation (Table 1), and center-crop models had higher shape bias than random-crop models throughout the training process (Appendix C.2). Further, texture bias decreased as minimum crop area used in random-crop augmentation increased (Appendix B).

**Appearance-modifying data augmentation reduces texture bias**. The majority of photographs in major computer vision datasets are taken in well-lit environments (94). Humans encounter much more varied illumination conditions. We hypothesized that the development of human-like shape representations might require more diverse training data than what is present in ImageNet. Geirhos et al. (91) previously showed that neural style transfer data augmentation induces a model to classify images according to shape. Here, we tested whether a similar effect could be achieved with more naturalistic forms of data augmentation that do not require training on stylized images, and that more closely resemble the variation encountered by humans. Following (13), we tested six individual augmentations as well as their compositions (see Appendix E.3 for details). We applied augmentations to a randomly selected 50% of the total examples in each mini-batch.

Table 3: **The effect of augmentations that reduce texture bias is additive.** ResNet-50 models were trained on ImageNet for 90 epochs with random-crop augmentation and augmentation p = 0.5 unless noted. Augmentations are cumulative across the rows (e.g. the "+ Gaussian blur" model used color distortion and Gaussian blur augmentation). "Stronger" augmentation used p = 0.75; "Longer" training is 270 epochs. The final two models are shape-biased (>50% shape-based classifications). Bolding indicates the highest value across augmentation types within a given column.

| Augmentation(s) | Shape Bias | Shape Match | Texture Match | ImageNet top-1 | top-5 | IN-Sketch top-1 | top-5 | SIN top-1 | top-5 |
|---|---|---|---|---|---|---|---|---|---|
| Baseline | 19.5% | 11.7% | **48.4%** | 76.6% | **93.3%** | 22.4% | 39.3% | 7.7% | 17.0% |
| + Color distortion | 25.8% | 15.3% | 44.2% | **76.9%** | **93.3%** | 28.1% | 46.6% | 9.9% | 20.5% |
| + Gaussian blur | 30.7% | 17.2% | 38.8% | 76.8% | **93.3%** | 29.0% | 47.9% | 11.1% | 21.9% |
| + Gaussian noise | 36.1% | 20.1% | 35.5% | 75.9% | 92.8% | 29.8% | 48.9% | 12.6% | 24.3% |
| + Min. crop of 64% | 48.7% | 29.1% | 30.7% | 73.5% | 91.5% | **30.9%** | **51.4%** | 14.5% | 28.2% |
| + Stronger aug. | 55.2% | 33.3% | 27.1% | 72.0% | 90.7% | 30.4% | 50.5% | **15.1%** | **28.8%** |
| + Longer training | **62.2%** | **38.3%** | 23.3% | 71.1% | 90.0% | 30.5% | 50.4% | 14.9% | 28.4% |

We found that many simple augmentations decreased texture bias relative to baseline (Table 2), and these effects were stronger when paired with center crops as opposed to random crops (Table D.1). In addition, as shown in Table 3, the effect of augmentations that reduce texture bias was additive. A model trained for 90 epochs with strong color distortion, Gaussian blur, Gaussian noise, and less aggressive random crop augmentation was actively shape-biased (55.2%), and one trained for longer (270 epochs) was even more shape-biased (62.2%). These results show that it is possible to induce a shape bias using naturalistic augmentation that does not include stylization. Augmentation that reduced texture bias according to the GST dataset also improved accuracy on the ImageNet-Sketch (IN-Sketch) (96) dataset, which consists of human-drawn sketches of each ImageNet class, and Stylized ImageNet (SIN) (36), which was generated by processing the ImageNet test set with a style transfer algorithm. These datasets privilege shape by ablating texture information in other ways.

In general, we observed a trade-off between ImageNet top-1 and shape bias (Figure 3). In Appendix A, we show that manipulating learning rate and weight decay during training produced a similar trade-off.

## 6 Effect of training objective

A hypothesis consistent with the results above is that texture bias is driven by the joint image-label statistics of the ImageNet dataset. To correctly label the many dog breeds in the dataset, for instance, a model would have to make texture-based distinctions between similarly shaped objects. To test this hypothesis, we compared the shape bias of models trained with standard supervised learning to models trained with self-supervised objectives different from supervised classification. To explore the interaction of objective and model architecture, we used each objective to train models with both AlexNet and ResNet-50 base architectures.

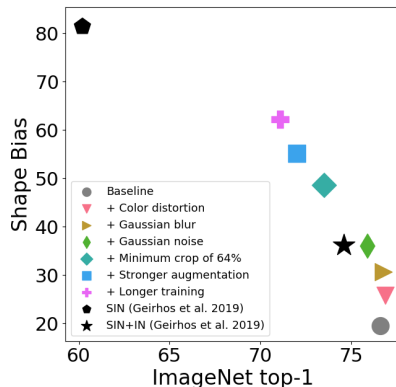

Figure 3: **Tradeoff between ImageNet top-1 and shape bias for models trained with different augmentation.** ResNet-50 models with naturalistic data augmentation achieve comparable tradeoffs to those from Geirhos et al. (36).

**Self-supervised losses.** We paired each self-supervised objective with up to two architecture backbones: AlexNet and ResNet-50 v2 (Appendix E.4.1), allowing for comparison with architecture-matched supervised counterparts.

*Rotation classification.* Input images are rotated 0, 90, 180, or 270 degrees, and the task is to predict which rotation was applied (chance = $1/4$) (39; 54). In their original presentation of this loss, Gidaris et al. (39) argued that to do well on this task, a model must understand which objects are present in an image, along with their location, pose, and semantic characteristics.

*Exemplar.* This objective, first introduced by (26), learns a representation where different augmentations of the same image are close in the embedding space. Our implementation follows (54). At

Table 4: **Both objective and base architecture affect shape bias.** We trained AlexNet and ResNet-50 base architectures on five objectives (rows). For SimCLR, we additionally include a baseline with the same augmentation. We froze the convolutional layers, reinitialized and retrained the fully connected layers, and evaluated the models using the GST stimuli. Shape match is generally higher for models with an AlexNet than ResNet-50 base architecture, while the reverse is true for texture. Rank order of shape bias across tasks is largely preserved across base architectures.

| Objective | Shape Bias | | Shape Match | | Texture Match | | ImageNet Top-1 Acc. | |
|---|---|---|---|---|---|---|---|---|
| | AlexNet | ResNet-50 | AlexNet | ResNet-50 | AlexNet | ResNet-50 | AlexNet | ResNet-50 |
| Supervised | 29.8% | 21.9% | 17.5% | 13.5% | 41.2% | 48.2% | 57.0% | 75.8% |
| Rotation | 47.0% | 32.3% | 21.6% | 14.2% | 24.3% | 29.8% | 44.8% | 44.4% |
| Exemplar | 29.9% | 14.4% | 12.6% | 7.5% | 29.5% | 44.7% | 37.2% | 41.8% |
| BigBiGAN | - | 31.9% | - | 17.7% | - | 37.7% | - | 55.4% |
| SimCLR | - | 37.0% | - | 17.3% | - | 29.4% | - | 69.2% |
| Supervised w/ SimCLR aug. | - | 40.4% | - | 23.1% | - | 34.0% | - | 76.3% |

training time, each batch consists of 8 copies each of 512 dataset examples with different augmentations (random crops from the original image, converted to grayscale with a probability of $2/3$). A triplet loss encourages distances between augmentations of the same example to be smaller than distances to other examples.

*BigBiGAN.* The BiGAN framework (24; 27) jointly learns a generator $\mathcal{G}(\mathbf{z})$ that converts latent codes $\mathbf{z}$ to images and an encoder $\mathcal{E}(\mathbf{x})$ that converts dataset images $\mathbf{x}$ to latent codes. At training time, the encoder and generator are optimized adversarially with a discriminator. The discriminator is optimized to distinguish pairs of sampled latents with their corresponding generator output $(\mathbf{z}, \mathcal{G}(\mathbf{z}))$ from pairs of dataset images with their corresponding latents $(\mathcal{E}(\mathbf{x}), \mathbf{x})$, whereas the generator and encoder are optimized to minimize the discriminator's performance. We use the representation from the penultimate layer of a ResNet-50 v2 encoder (25).

*SimCLR.* This model learns representations by maximizing agreement between differently augmented views of the same data example via a contrastive loss in the latent space. SimCLR uses a composition of augmentations, namely random crop, color distortion, and Gaussian blur, and leverages a non-linear transformation (projection head) to transform the representation before applying the contrastive loss (based on softmax cross-entropy). We follow the same setting as in (13), using the ResNet-50 base architecture, and training the model for 1000 epochs with a batch size of 4096. To test the contribution of augmentation alone, we created a supervised baseline with the same augmentation and training ("Supervised w/ SimCLR aug." in Table 4). See also Appendix E.4.1.

**Evaluation of shape bias.** To facilitate comparison across supervised and self-supervised tasks, we trained classifiers on top of the learned representations using the standard ImageNet dataset and softmax cross-entropy objective, freezing all convolution layers and reinitializing and retraining later layers. We provide full training details in Appendix E.4.2. See also Appendix Tables D.2 and D.3.

**Results.** We found effects of both objective and base architecture on texture bias (Table 4). The rotation model had significantly lower texture bias than supervised models for both architectures; this may be because rotationally-invariant texture features are not useful for performing the rotation classification task. In general, shape match was higher for models with AlexNet than ResNet-50 architecture (log odds 1.04, 95% CI [0.83, 1.24], logistic regression; see Appendix E.4.3 for details), whereas the reverse was true for texture match (log odds 0.78, 95% CI [0.67, 0.92]). The effects of architecture and task appear to be largely independent: the rank order of shape bias across tasks was similar for the two model architectures. Amongst the unsupervised ResNet-50 models, SimCLR had the lowest texture bias. However, a supervised baseline with the same augmentation had a still-lower texture bias, suggesting that augmentation rather than objective was the more important factor.

# 7   Effect of architecture

**Shape bias and ImageNet accuracy.** Figure 4 shows the shape bias, shape match, and texture match of 16 high-performing ImageNet models trained with the same hyperparameters (Appendix E.5.1 for details). Both shape bias and shape match correlated with ImageNet top-1 accuracy, whereas texture match had no significant relationship. These results suggest that models selected for high ImageNet

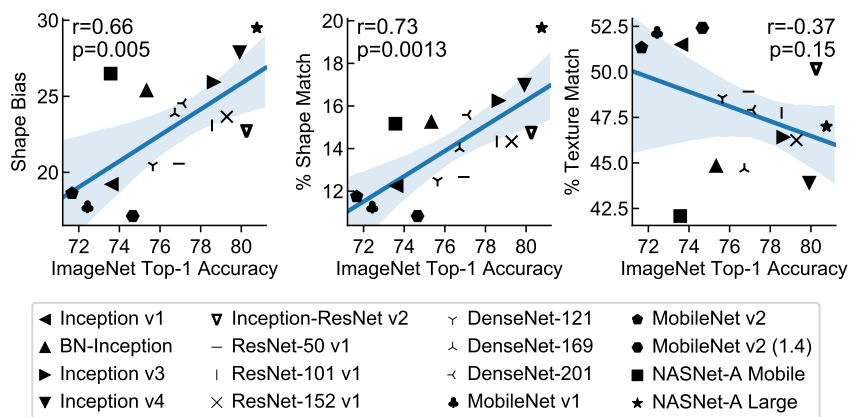

Figure 4: **Among high-performing ImageNet models, shape bias and accuracy correlate with ImageNet accuracy.** $p$-values indicate significance according to a $t$ distribution. Blue line is a least squares fit to the plotted points; shaded area reflects 95% bootstrap confidence interval.

top-1 are more effective at extracting shape information. However, the asymmetry with Figure 3 shows that models selected for high shape bias do not necessarily have high ImageNet top-1.

**Shape bias in neurally motivated models.** Human visual judgments are well known to be shape-biased. Would a model explicitly built to match the primate visual system display lower texture bias than standard CNNs? We tested the shape bias of the CORNet models introduced by Kubilius et al. These models have architectures specifically designed to better match the primate ventral visual pathway both structurally (depth, presence of recurrent and skip connections, etc.) and functionally (behavioral and neural measurements) (59). We computed the shape bias of CORNet-Z, -R, and -S, using the publicly available trained models (18). The simplest of these models, CORNet-Z, had a shape bias of 14.9%, shape match of 9.2%, and texture match of 52.2%. CORNet-R, which incorporates recurrent connections, had a shape bias of 36.7%, and shape and texture accuracies of 19.6% and 33.8%, respectively. CORNet-S, the model with the highest BrainScore (81), had a shape bias of 20.3%, and shape and texture accuracies of 13.3% and 51.9%. Taken together, these models did not exhibit a greater shape bias than other models we tested. Perhaps surprisingly, the texture match was in the high range of those we observed.

**Shape bias of attention vs. convolution.** We wondered whether convolution itself could be a cause of texture bias. Ramachandran et al. (75) recently proposed a novel approach to image classification that replaces every convolutional layer in ResNet-50 v1 with local attention, where attention weights are determined based on both relative spatial position and content. We compared this model (with spatial extent $k = 7$, and 8 attention heads) against a baseline ResNet-50 v1 model trained with the same hyperparameters. The attention model had a shape bias of 20.2% (shape match: 12.8%; texture match: 50.7%), similar to the baseline's shape bias of 23.2% (shape match: 14.4%; texture match: 47.7%). Thus, use of attention in place of convolution appears to have little effect upon texture bias.

## 8 Degree of representation of shape and texture in ImageNet models

Standard ImageNet-trained CNNs are biased towards texture in their classification decisions, but this does not rule out the possibility that shape information is still represented in layers of the model prior to the output. We tested the extent to which it is possible to decode stimulus shape versus texture in the final layers of AlexNet and ResNet-50. We trained linear multinomial logistic regression classifiers on two classification tasks. Taking as input activations from a layer of a frozen, ImageNet-trained model, the classifier predicted either (i) the shape of a GST image or (ii) its texture. We used center-crop AlexNet and ResNet-50, considering AlexNet "pool3" (final convolutional layer, including the max pool), "fc6" (first linear layer of the classifier, including the ReLU), and "fc7" (second linear layer), and ResNet-50 "pre-pool" (output of the final bottleneck layer) and "post-pool" (following the global average pool). See Section E.6 for additional details, and Figure C.4 for decoding results from all AlexNet layers.

**Results.** We found that, despite the high texture bias of AlexNet and ResNet-50, shape could nonetheless be decoded with high accuracy (Figures 5 and C.4). In fact, it was possible to classify shape (77.9%) more accurately than texture (65.6%) throughout the convolutional layers of AlexNet. In ResNet-50, while decoding accuracy for texture (80.9%) was higher than for shape (66.2%) for pre-pool, shape accuracy was still high. Interestingly, shape accuracy decreased through the fully-connected layers of AlexNet's classifier, and from ResNet pre-pool to post-pool, suggesting that these models' classification layers remove shape information.

## 9 Conclusion

From the perspective of the computer vision practitioner, our results indicate that models that prefer to classify images by shape rather than texture outperform baselines on some out-of-distribution test sets (IN-Sketch and SIN). We suggest practical ways to reduce texture bias, for example using additive augmentation (color distortion, blur, noise) and center crops. At the same time, we show that there is often a trade-off between shape bias and ImageNet top-1 accuracy.

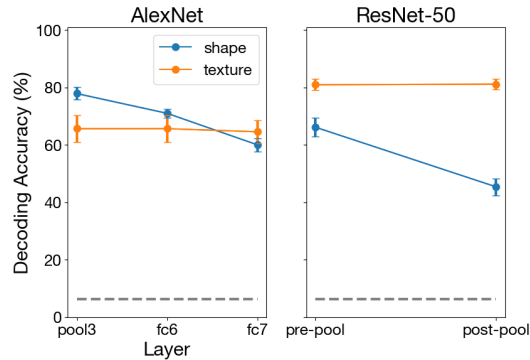

Figure 5: **Both the shape and texture of ambiguous images is decodable from the hidden representations of texture-biased ImageNet-trained models.** Performance of linear classifiers trained to classify shape (blue) or texture (orange) of the GST stimuli given layer activations from frozen ImageNet-trained AlexNet (left) and ResNet-50 (right). Performance is the maximum classification accuracy over the training period (mean ± SD across 5 splits). Chance is 6.25%. At the final convolutional layer, both models contain substantial shape information, but it decreases subsequently.

On the scientific side, our findings flesh out the picture of how and why ImageNet-trained CNNs may differ from human vision. While both architecture and training objective have an effect on the level of texture bias in a model, the statistics of the training dataset are the most important factor. Changing these statistics using data augmentations qualitatively similar to those induced by the human visual system and visual environment is the most effective way to instill in CNNs shape-biased representations like those documented in the human psychological literature.

## Broader Impact

People who build and interact with tools for computer vision, especially those without extensive training in machine learning, often have a mental model of computer vision models as similar to human vision. Our findings contribute to a body of work showing that this view is actually far from correct, especially for ImageNet, one of the datasets most commonly used to train and evaluate models. Divergences between human and machine vision of the kind we study could cause users to make significant errors in anticipating and reasoning about the behavior of computer vision systems. Our findings contribute to a body of work delineating divergences between human and machine vision, and suggesting avenues for bringing the two systems closer together. Allowing people from a wide range of backgrounds to make safe, predictable, and equitable models requires vision systems to perform at least roughly in accordance with their expectations. Making computer vision models that share the same inductive biases as humans is an important step towards this goal. At the same time, we recognize the possible negative consequences of blindly constraining models' judgments to agree with people's: human visual judgments display forms of bias that should be kept out of computer models. More broadly, we believe that work like ours can have a beneficial impact on the internal sociology of the machine learning community. By identifying connections to developmental psychology and neuroscience, we hope to enhance interdisciplinary connections across fields, and to encourage people with a broader range of training and backgrounds to participate in machine learning research.

## Acknowledgments and Disclosure of Funding

We thank Jay McClelland, Andrew Lampinen, Akshay Jagadeesh, and Chengxu Zhuang for useful conversations, and Guodong Zhang and Lala Li for comments on an earlier version of the manuscript. KLH was supported by NSF GRFP grant DGE-1656518.

## Footnotes

[1]Note that in the experiments detailed here, we trained models to classify ambiguous images, using either shape or texture labels as targets. In the other experiments in the paper, we trained models on ordinary ImageNet and evaluated shape bias on ambiguous images.

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
