[Supplementary Material]

# Supplementary Material for "The Origins and Prevalence of Texture Bias in Convolutional Neural Networks"

## A    Hyperparameters that maximize accuracy do not maximize shape bias

Previous work has found that in addition to model architecture, training procedure details are an important determiner of the representations a model ends up learning (31; 97; 55; 67; 63; 68; 102). Extracting optimal performance from neural networks requires tuning hyperparameters for the specific architecture and dataset. However, the hyperparameters that optimize performance on a held-out validation set drawn from the same distribution do not necessarily optimize for either shape or texture bias. In order to determine whether there were consistent patterns in the relationship between hyperparameters and shape or texture match, we performed a hyperparameter sweep across a grid of learning rate and weight decay settings. We trained ResNet-50 (2) networks on the 16 ImageNet superclasses used by Geirhos et al. (36), taking 1000 images from each superclass, and computed mean per-class accuracy on the corresponding ImageNet validation subset. We trained networks for 40,000 steps at a batch size of 256 using SGD with momentum of 0.9 with a cosine decay learning rate schedule and standard random-crop preprocessing, and averaged results across 3 runs.

Figure A.1: **Higher learning rates produce greater shape bias.** Plots show mean of 3 runs on 16-class ImageNet. Results for hyperparameter combinations achieving <70% accuracy are masked. We plot weight decay × learning rate because it is more closely related to accuracy than weight decay (56; 14).

As shown in Figure A.1, higher values of learning rate and weight decay were associated with greater shape match and shape bias, whereas lower learning rates were associated with greater texture match. We observed the highest shape match at the highest learning rate where the network could be reliably trained and the highest texture match at the lowest learning rate tested. Mean per-class accuracy on 16-class ImageNet was sensitive to the value of the product of the weight decay and learning rate, but relatively insensitive to the value of the learning rate itself.

We hypothesize that these hyperparameters may have a similar effect on training to the forms of stochastic data augmentation investigated in Section 5. Training with higher learning rates introduces more minibatch noise into the training process, which may have a similar effect to adding noise to the input.

# B   Increasing random crop area reduces texture bias

We found that random-crop augmentation biases models towards texture (Section 5). To further understand the relationship between random-crop augmentation and texture bias, we varied the proportion of the image covered by the random crops taken at training time. In standard ImageNet data augmentation, the area of the random crop is sampled uniformly from $[0.08, 1.0]$. Here we increased the minimum area to 0.16, 0.32, 0.48 and 0.64. With larger minimum area, the average crop area increases. We did not change the aspect ratio or other data augmentation settings. We observed that increasing the minimum crop area helps reduce texture bias but also reduces the ImageNet top-1 accuracy.

Figure B.1: **Shape bias increases as a function of minimum crop area, while ImageNet top-1 accuracy decreases.** Shape bias versus ImageNet top-1 for ResNet-50.

# C Supplemental Figures

Figure C.1: **Networks with limited receptive fields learn texture more easily than shape.** Data efficiency (top row) and learning dynamics (bottom row) for BagNet-17, a model that makes classifications based on local ($17 \times 17$ pixel) image patches without considering their spatial configuration (11), achieved higher texture than shape accuracy on both the GST and Navon datasets.

Figure C.2: **Random-crop preprocessing results in greater texture bias throughout training.** Plot shows ImageNet top-1 validation accuracy, as well as shape match, texture match, and shape bias on the GST dataset, for Inception-ResNet v2 models trained with random-crop (solid) and center-crop (dashed) preprocessing. Although the model trained with center crops achieved substantially lower peak ImageNet top-1 validation accuracy compared to the model trained with random crops (77.3% vs. 80.3%), it achieved higher shape match at all stages of training. For both models, texture match reached its peak very early in training and dropped as training continued.

Figure C.3: Shape match, texture match, and shape bias by shape class (shape match, shape bias) or texture class (texture match) label on the GST dataset for AlexNet (blue), VGG16 (orange), and ResNet-50 (green) models trained on ImageNet with random-crop (top row) or center-crop (bottom row) preprocessing.

Figure C.4: **Shape is persistently more decodable through the convolutional layers of AlexNet than is texture, which rises through them. In the fully connected layers, shape decodability decreases, whereas texture increases.** Performance of linear classifiers trained to classify the shape or texture of GST stimuli given layer activations (including the ReLU for convolutional layers) from the frozen ImageNet-trained AlexNet whose results also appear in Figure 5).

# D  Supplemental Tables

Table D.1: **Color distortion, Gaussian blur, Gaussian noise, and Sobel filtering reduce texture bias in center-crop models.** ResNet-50 models were trained on ImageNet with center crops and without random flips for 90 epochs. Augmentations were applied with 50% probability
.

| Augmentation | Shape Bias | Shape Match | Texture Match | ImageNet Top-1 Acc. |
|---|---|---|---|---|
| Baseline | 25.2% | 15.2% | 45.1% | 69.7% |
| Rotate 90°, 180°, 270° | 19.1% | 11.3% | 47.9% | 70.7% |
| Cutout | 20.3% | 12.1% | 47.4% | 71.5% |
| Sobel filtering | 25.1% | 14.4% | 42.9% | 52.4% |
| Gaussian blur | 29.6% | 17.4% | 41.5% | 68.6% |
| Color distort. | 33.3% | 19.9% | 39.9% | 69.9% |
| Gaussian noise | 42.2% | 23.2% | 31.7% | 67.7% |

Table D.2: **Linear and nonlinear classifiers trained on self-supervised AlexNet representations show similar shape bias.** We trained multinomial logistic regression classifiers to classify ImageNet based on the representation of the final AlexNet pooling layer ("pool3"). Although these classifiers perform worse on ImageNet compared to freezing the convolutional layers and retraining the three layer MLP at the end of the AlexNet architecture, performance on the GST dataset is similar.

| Objective | Shape Bias | | Shape Match | | Texture Match | | ImageNet Top-1 Acc. | |
|---|---|---|---|---|---|---|---|---|
| | pool3 | MLP | pool3 | MLP | pool3 | MLP | pool3 | MLP |
| Supervised | 33.2% | 29.8% | 18.2% | 17.5% | 36.5% | 41.2% | 53.1% | 57.0% |
| Rotation | 51.1% | 47.0% | 20.6% | 21.6% | 19.7% | 24.3% | 36.6% | 44.8% |
| Exemplar | 31.7% | 29.9% | 12.7% | 12.6% | 27.4% | 29.5% | 32.5% | 37.2% |

Table D.3: $k$-**nearest neighbors evaluation of self-supervised representations**. Following the same procedure as described in section 8, we trained k-Nearest Neighbors classifiers (k=5 with Euclidean distance; Sklearn implementation) to classify GST images from the layer activations of ImageNet-trained models, using layer pool3 for AlexNet and the penultimate layer for ResNet-50 v2. To classify a given item (e.g. a knife-elephant), we relabeled each other item in the dataset as a "shape match" (e.g. shape = knife), "texture match" (texture = elephant), "both match" (other images of elephant-texture knives), or "other" (e.g. a cat-bottle). We excluded from this set of options items that had been generated through the neural style transfer process using the same content and/or style target images as the test item. Table values indicate the percentage of the dataset assigned each label type.

| Objective | Shape Match | | Texture Match | | Both Match | | Other | |
|---|---|---|---|---|---|---|---|---|
| | AlexNet | ResNet-50 | AlexNet | ResNet-50 | AlexNet | ResNet-50 | AlexNet | ResNet-50 |
| Supervised | 10.2% | 4.3% | 12.6% | 60.5% | 1.4% | 2.2% | 75.8% | 33.0% |
| Rotation | 13.4% | 3.3% | 10.2% | 42.7% | 1.8% | 1.0% | 74.7% | 53.0% |
| Exemplar | 3.6% | 0.3% | 30.6% | 40.8% | 1.6% | 0.3% | 64.3% | 58.6% |
| BigBiGAN | - | 15.9% | - | 34.9% | - | 4.6% | - | 44.6% |

# E Methods

## E.1 Learning experiments

### E.1.1 Dataset considerations

On their own, none of these datasets are unproblematic representations of texture. For the GST dataset, for example, human subjects were unable to attain good performance on the style classification task (mean accuracy = 14.2%, chance = 6.25%; analysis of data from (36) human experiment in which subjects were given texture-biased instructions, originally presented in Fig 10b of (36) plotted by shape class; data obtained from (91)). Further, the performance of the style transfer algorithm on individual images introduces another source of variability, and the fact that style transfer itself relies on ImageNet-trained CNN features means that the data were not generated independently of the models being evaluated. The Navon stimuli, meanwhile, strongly deviate from the statistics of natural images. Finally, the noise textures from ImageNet-C arguably deviate the farthest from what people generally mean by the term. We hope that presenting results for all three datasets will dilute any idiosyncracies of the datasets individually. In future work, we hope to create new datasets that combine the controllability of the Navon stimuli with the naturalism of the GST and ImageNet-C datasets.

As shown in Figure C.1, we found that BagNet-17, a model that makes classifications based on local image patches without considering their spatial configuration (11), was able to classify shape less well than texture for both the GST and Navon datasets, suggesting it is necessary to use global features to classify shape for this dataset.

### E.1.2 Dataset splits

*Geirhos Style-Transfer (GST) dataset*. We created 5 cross-validation splits of the data, using each cv split for both classification tasks. To create a given split, we held out a single shape exemplar and a single texture exemplar, and confirmed that no whole shape or texture classes were held out. During the texture task, then, a model was required to generalize a given texture across exemplars of that texture; during the shape task, it had to generalize a given shape across exemplars of that shape. The mean validation size over cv splits was 483 items (40.3% of the data). Although the dataset contains 80 images where shape and texture match, (36) excluded these when computing shape and texture bias, and we exclude these from our experiments.

*Navon dataset*. For the Navon dataset, we created 5 cv splits independently for each task. For the shape task, we held out 3 texture classes (e.g. the letters "T", "U", "E"), and for the texture task, we held out 3 shape classes. The validation size was 375 items (11.5% of the data).

*ImageNet-C dataset*. We split each version of the dataset separately for the shape and texture tasks. For the shape task, we held out 2 texture classes (e.g. "snow", "fog"); for the texture task, we held out two shape classes (e.g. wnid's "n01632777", "n03188531"). The validation size was 9,500 items (10.5% of the data).

### E.1.3 Training

We trained AlexNet models with the output layer modified to reflect the number of classes present in the dataset at hand (GST: 16, Navon: 26, ImageNet-C: 19). We additionally reduced the widths of the fully connected layers in proportion to the reduction in number of output classes vs. ImageNet.

We preprocessed training and validation images by normalizing the pixel values by the mean and standard deviation of the subset of data used for training. For the GST and ImageNet-C datasets, we randomly horizontally flipped each training image with p = 0.5 during training.

In subsampling the training data, we enforced the condition that an instance of each shape and texture class appear at least once. For the GST dataset, we held out one texture and one shape exemplar in each split; for other datasets, we held out shape classes when training on texture and texture classes when training on shape. We trained all models for 90 epochs using Adam (53) with a learning rate of $3 \times 10^{-4}$, weight decay of $10^{-4}$, and batch size of 64.

## E.2 Evaluation of shape bias, shape match, and texture match

To evaluate shape and texture match and shape bias in ImageNet-trained models, following (36), we presented models with full, uncropped images from the GST dataset, collected the class probabilities returned by the model, and mapped these to the 16 superclasses defined by (36) by summing over the probabilities for the ImageNet classes belonging to each superclass. Shape match was the percentage of the time a model correctly predicted probe items' shapes, texture match was the percentage of the time the model correctly predicted probe items' textures, and shape bias was the percentage of the time the model predicted shape for trials on which either shape or texture prediction was correct.

## E.3 Data augmentation experiments

### E.3.1 Models

*AlexNet, VGG16.* We used implementations available through torchvision (`https://github.com/pytorch/vision`). We trained these models for 90 epochs using SGD with a momentum of 0.9, an initial learning rate of 0.0025, and a batch size of 64, and with weight decay of $10^{-4}$. We decayed the learning rate by a factor of 10 at epochs 30 and 60. We evaluated shape bias and shape and texture match at the point over the training period corresponding to maximum classification accuracy on the validation set.

*ResNet-50.* We used the ResNet-50 implementation available as part of the SimCLR (13) opensource code (85). We trained the model for 90 epochs using SGD with a momentum of 0.9 at a batch size of 4096. A cosine learning rate decay was used similar to (13).

*Inception-ResNet v2.* We used the implementation from TensorFlow-Slim (`https://github.com/tensorflow/models/tree/master/research/slim`), trained as described in Section E.5.1. To evaluate shape and texture match, we used the checkpoint that achieved the highest accuracy on the ImageNet validation set, at 122 epochs with random crops and 58 epochs without random crops.

Our results for AlexNet and VGG16 differ slightly from those in Geirhos et al. (36), which reported shape biases of AlexNet (42.9%) and VGG16 (17.2%) models implemented in Caffe. Since publication of their paper, they have reported the shape biases for PyTorch implementations of these models, with the random-crop preprocessing we have described, obtaining 25.3% for AlexNet, 9.2% for VGG16, 22.1% for ResNet-50 (91). Using the pretrained models available through PyTorch's model zoo, which uses random-crop preprocessing, we obtained comparable results to theirs: 26.9% for AlexNet, 10% for VGG16, and 22.1% for ResNet-50; slight differences may be due differences in random initialization. These models were trained with a batch size of 256, but the results we report in Table 1 and Figure C.3 for our models trained at batch size 64 are within a few percentage-points of the larger-batch-size models.

### E.3.2 Augmentation operators

For all experiments presented in Section 5 (Tables 1, 2, 3; Figure 3), we used random-flip augmentation. In addition, we tested the effects of color distortion (color jitter with probability of 80% and color drop with probability of 20%), rotation, cutout, Gaussian noise, Gaussian blur (kernel size was 10% of the image width/height), Gaussian noise, and Sobel filtering, all as specified by (13). Unless otherwise noted, we applied augmentations to each example with a probability of 50% (which is equivalent to randomly selected roughly 50% of the total examples in each mini-batch of 4096 items). An illustration of these augmentations, reproduced by permission of (13), appears in Figure E.1.

## E.4 Self-supervised representation experiments

### E.4.1 Self-supervised training

*AlexNet.* We trained AlexNet models from scratch using a modified version of the code provided by Kolesnikov et al. (54). Unlike AlexNet models used for other experiments, these models were trained using TensorFlow rather than PyTorch, and thus the shape and texture bias of the baseline supervised model are slightly different. As the base network, we used the AlexNet implementation from TensorFlow-Slim (`https://github.com/tensorflow/models/tree/`

(a) Original      (b) Crop, resize (and flip)      (c) Color distort. (drop)

(d) Color distort. (jitter)      (e) Rotate $\{90°, 180°, 270°\}$      (f) Cutout

(g) Gaussian noise      (h) Gaussian blur      (i) Sobel filtering

Figure E.1: Illustrations of the studied data augmentation operators. Each augmentation can transform data stochastically with some internal parameters (e.g. rotation degree, noise level). Reproduced by author permission from (13), which contains detailed descriptions of augmentation operations.

master/research/slim). For consistency with the PyTorch AlexNet, we modified the first convolutional layer of the TensorFlow-Slim network to use padding and trained at $224\times 224$ pixel resolution. Unlike Gidaris et al. (39), we did not use batch normalization. We trained all AlexNet models for 90 epochs using SGD with momentum of 0.9 at a batch size of 512 examples, with a weight decay of $1e-4$ and an initial learning rate of 0.02. We decayed the learning rate by a factor of 10 at epochs 30 and 60. For all models, we used preprocessing consisting of random crops sampled as random proportions of the original image size and random flips.

*ResNet-50 v2.* For rotation, exemplar, and supervised losses, we used ResNet-50 v2 models made available as part of the Visual Task Adaptation Benchmark (104; 95). For BigBiGAN, we used the public model (21).

*SimCLR* and *ResNet-50 w/ SimCLR augmentation.* For SimCLR and ResNet-50 w/ SimCLR augmentation (Table 4), we used models made available as part of the SimCLR (13) open-source code (85).

### E.4.2 Training supervised classifiers on self-supervised representations

We trained all classifiers using SGD with momentum of 0.9 with data augmentation consisting of random flips and random crops obtained by resizing the image to 256 pixels on its shortest side and cropping $224 \times 224$ regions. This less aggressive form of cropping was used for training classifiers

on top of self-supervised representations in previous work (39; 54; 25), and we found it to be essential to produce their results.

For fair comparison between supervised and self-supervised models, Table 4 presents supervised AlexNet results where the network was first trained with aggressive random crops sampled as random proportions of the original size and random flips, and then the convolutional layers were frozen and the fully connected layers were retrained with the less aggressive cropping strategy described above, thus replicating the training procedure for the self-supervised AlexNet models. However, the model obtained by retraining the fully connected layers performed very similarly to the original model. Both obtained ImageNet top-1 accuracies of 57.0%, and shape bias was also nearly identical (original model: 30.6%; model with retrained fully connected layers: 29.9%).

*Logistic regression on ResNet representations.* We trained for 520 epochs at a batch size of 2048 and an initial learning rate of 0.8 without weight decay, decaying the learning rate by a factor of 10 at 480 and 500 epochs.

*AlexNet MLP training.* When retraining the MLP at the end of AlexNet networks, we trained for 90 epochs at a batch size of 512 with an initial learning rate of 0.02. decayed by a factor of 10 at 30 and 90 epochs. We optimized weight decay by choosing the best value out of $\{10^{-3}, 10^{-4}, 10^{-5}, 10^{-6}\}$ on a held-out set of 50,046 examples, and then trained on the full ImageNet dataset. Optimal values for weight decay were $10^{-3}$ for the supervised model, $10^{-4}$ for rotation, and $10^{-5}$ for the exemplar loss.

*Logistic regression on AlexNet pool3 layer.* We trained for 600 epochs, decaying the learning rate by a factor of 10 at 300, 400, and 500 epochs. As for AlexNet MLP training, we optimized weight decay on a held out validation set. The optimal values did not change.

### E.4.3   Statistical modeling

We performed statistical modeling of the effects of self-supervised loss and architecture using logistic regression. We modeled the logit of the probability of correct shape/texture classification of each example with each network as a linear combination of effects of architecture, loss, the individual example, and an intercept term. This model is a generalization of repeated measures ANOVA where the dependent variable is binary. We fit the model using iteratively reweighted least squares using statsmodels (82). We excluded examples that all networks classified correctly or incorrectly; these do not affect the values of parameters corresponding to architecture or loss, but cause per-example parameters to diverge during model fitting. Coefficients provided in the paper are maximum likelihood estimates with Wald confidence intervals computed based on the corresponding standard errors from the Fisher information matrix.

### E.5   Architecture experiments

### E.5.1   Training settings for comparison of ImageNet architectures

We trained at a batch size of 4096 using SGD with Nesterov momentum of 0.9 and weight decay of $8 \times 10^{-5}$ and performed evaluation using an exponential moving average of the training weights computed with decay factor 0.9999. The learning rate schedule consisted of 10 epochs of linear warmup to a maximum learning rate of 1.6, followed by exponential decay at a rate of 0.975 per epoch. For all conditions we randomly horizontally flipped images and performed standard Inception-style color augmentation.

### E.6   Decoding experiments

We decoded from the center-crop AlexNet that appears in Table 1, and from a center-crop ResNet-50 model implemented in torchvision (https://github.com/pytorch/vision) and trained for 90 epochs using SGD with momentum of 0.9 at a batch size of 64 and with weight decay of $10^{-4}$. The initial learning rate was 0.025, which we decayed by a factor of 10 at epochs 30 and 60. We randomly horizontally flipped training images. This model had a shape bias of 25.9%, shape match of 15.7%, texture match of 44.9%, and ImageNet top-1 accuracy of 70.6%.

We trained linear classifiers to classify either the shape or texture of the GST images given activations from some model layer. For each model layer-task pair, we first found a learning rate that effectively

optimized the classifier, then searched over weight decay settings. We evaluated the mean classification accuracy for classifiers trained separately on each of the 5 splits of the data described in E.1.2. We trained each classifier for 90 epochs using Adam (53) at batch size 64.