[Reviews · NeurIPS 2020]

Review 1

Summary and Contributions: This paper works to determine the factors that cause current ImageNet-trained CNNs to be biased towards texture. The successfully isolate several factors, and additionally evaluate the bias of non-supervised methods.

Strengths: This is the first principled analysis I know of investigating the phenomenon of texture bias. The writing is very clear, and I found the analysis quite insightful. Overall, it is a very engaging and interesting read. The findings of aggressive random cropping exacerbating texture bias are extremely interesting. I feel that the results of Appendix B could merit inclusion in the main body of the paper. The finding that SimCLR is more biased towards shape, but then recognizing that this is due to the augmentation and then applying said augmentation to supervision is clever.

Weaknesses: The figures generally need much improvement Figure 1: WordNet IDs should be text labels Figure 2: Aesthetically this figure looks very low quality despite the information contained. Error bars shouldn't be the same width as the lines, font is non-standard. Having AlexNet and ResNet50 on different panels makes them hard to compare. I would recommend plotting both architectures on the same subplot Figure 3: The random shape sequence is not very informative. I would recommend color coding the sequence in a rainbow. Font size should be increased for the legend. Figure 4: The font is small enough to be hard to read (overall I do like this finding though) Figure 5: Stylistically many problems are shared with Figure 2. In general Tables besides 1 could have more guiding visuals (e.g. bolding)

Correctness: Yes, I find the experiments and claims very sound.

Clarity: The text, especially the higher-level sections, is generally well-written and engaging, but as previously discussed the presentation of results is not as high quality.

Relation to Prior Work: The necessary citations are there, but sometimes the authors cite a long list of publications with insufficient context (L23, 85). These should be split into several chunks with distinctions/relevance clearly explained.

Reproducibility: Yes

Additional Feedback: I find your experiments extremely interesting with good results and insight, but unfortunately I don't think the current state of the presentation is publication-quality. I'd be interested in a comparison of your architecture findings to those of Ilyas et al.[1] In Figure 5, I would like to see intermediate (not after the final conv) layers included. Shape vs texture results would be interesting to contrast with the generic classification results of [2] when examining intermediate layers. Why did you opt for Exemplar as one of your methods in S6? It is fairly dated, especially if you're running it alongside of SimCLR. It seems to me like Instance Discrimination would be a better choice if you wanted a contrastive baseline to compare to SimCLR. The findings on BigBiGan are very interesting, but I think baselines of purely generative and encoder-based methods should be included as ablation (e.g. GAN and VAE) The findings of S7 Paragraph2 should be presented in a small table or figure rather than simply listing numbers. Similar comments for Paragraph3. Typo?: L254, "do not necessarily"? L307/8: Which experiments are you referring to as OOD? [1] Ilyas, Andrew, Shibani Santurkar, Dimitris Tsipras, Logan Engstrom, Brandon Tran, and Aleksander Madry. "Adversarial examples are not bugs, they are features." In Advances in Neural Information Processing Systems, pp. 125-136. 2019. [2] Asano, Yuki M., Christian Rupprecht, and Andrea Vedaldi. "A critical analysis of self-supervision, or what we can learn from a single image." arXiv preprint arXiv:1904.13132 (2019). UPDATE AFTER REBUTTAL: As the author's addressed my concerns concerning the visual presentation, I am happy to recommend acceptance (8) for this paper as well. The experiments and discussion are extremely well done and updated figures will make for an extremely complete and high quality work.


Review 2

Summary and Contributions: The paper presents an in-depth study of texture bias in CNNs. The authors use three datasets, the cue conflict dataset from Geirhos et. al. 2019, the Navon Dataset and ImageNet-C, and measure the effect of a variety of factors including the effect of data augmentation, self-supervised training objectives and model architecture. In my opinion there are two central findings: 1. CNNs can learn to classify based on shape equally well as on texture 2. Data augmentation during training plays a central role determining if a model is biased towards texture or shape. There is a wealth of additional findings including: - the effect of augmentations increasing the shape bias is cumulative - improving the shape bias is often at odds with ImageNet accuracy - self-supervised training does not by itself create models biased towards shape - shape bias somewhat correlates with accuracy in high-performing ImageNet models - shape information is mostly discarded in the last layers - innocent hyperparameters like random cropping, crop size and learning rate influence the bias and a number of more subtle points. Especially the experiment showing the additive effect of many data augmentation techniques and how they negatively influence ImageNet accuracy is extremely interesting. If a number of data augmentations can flip the model to having a shape bias but harms ImageNet accuracy that indicates that ImageNet may indeed not be suited to learn object recognition but rather encourages models to use texture and other subtle whole image cues that don’t require identifying the foreground. This seems to be very much in line with the results found in Kolesnikov et. al. 2019 Appendix G and if this effect is true it has the potential to shift the way the community vies ImageNet.

Strengths: Without a doubt this is the most in-depth study on shape and texture bias in CNNs and why they arise. There is a huge amount of insides in this paper and almost nothing that’s not to like.

Weaknesses: There are very few weaknesses, most of which I’d simply love to see because the work got me excited for the topic rather than things that are missing. The one thing that comes closes to an issue is with the self-supervised representation experiment specifically with the fine-tuning step. As the authors have established the shape bias of a model is strongly influenced by the data augmentation during training. However the data augmentation during fine-tuning is standard random flipping and cropping. As a result the model will have a similar bias as the standard model if we assume that the representation before the classifier contains both, texture and shape information, which is to be expected from the experiments in Section 8. In my opinion it would be interesting to either perform the fine-tuning with the same data augmentations as in the cumulative experiment or add the self-supervised representations to the decoding experiment in Section 8. To say it again there is nothing wrong with the current experiment and we definitely learn something from it: simply switching to self-supervised pre-training does not lead to models with a shape bias. There is however more insight to be gained from this. Regarding the things I’d love to see I think a study of newer models that were trained with different data augmentation strategies (e.g. AdvProp, mixup, AugMix, ...), newer models (FixResNet, EfficientNet, …) and probably most interesting additional data (BiT, ResNeXt-wsl, NoisyStudent, ...). But that is more of a fully-fledged follow-up than something missing from the submission.

Correctness: From what I can judge all claims and methods appear to be correct.

Clarity: I found the paper very clearly written and easy to understand.

Relation to Prior Work: Yes, the paper does a very good job at discussing and structuring prior work. The difference between the prior work and the presented study are absolutely clear.

Reproducibility: Yes

Additional Feedback: References: Geirhos et. al. 2019: ImageNet-trained CNNs are biased towards texture; increasing shape bias improves accuracy and robustness Kolesnikov et. al. 2019: Big Transfer (BiT): General Visual Representation Learning On the broader impact section: I miss any discussion of the negative impact this work can have or rather that some of the main findings imply. Most of the broader impact statement discusses how this study can help identifying and closing the gap between human and machine vision but as of now the main finding is that performance on ImageNet, the benchmark we as a field rely on most, is to some extend anticorrelated with a shape bias and thus anti-correlated with a strong bias of human perception. This is definitely worrisome, and I miss that aspect to be clearly stated in the broader impact section. Right now, it reads more like an extended outlook advertising all the positive future impact this work could have. I agree with this potential impact but at least for me listing only this is not what I expect from a broader impact statement. ### Post-rebuttal comment: I thank the authors for updating the broader impact statement. The proposed addition sounds good and fixes my issue. ### General Post-rebuttal comment: Overall the rebuttal and the other reviews did not change my assessment that this is a good paper and should be accepted. I think it is noteworthy that the authors promised to extend section 5 with results for all layers and generally implemented a number of suggestions all strengthening the paper.


Review 3

Summary and Contributions: This paper offers a more detailed inspection of the popular claim that "ImageNet-trained CNNs are biased towards texture" and studies the underlying reason of such observation, leading to the conclusion that the differences "arise not from differences in their internal workings, but from differences in the data that they see." Then the paper answers multiple related questions regarding this central conclusion, such as how different self-supervised loss (i.e. regularizations) play different roles in learning the bias, and how different augmentation strategies play different roles.

Strengths: The paper contributes to an important research question generally regarding the learning behavior of convolutional neural networks, it is built upon massive amount of work and answer many questions thoroughly.

Weaknesses: Some questions are not answered well. For example, the results in Section 8 do not look very convincing (See details later).

Correctness: I believe so.

Clarity: Yes.

Relation to Prior Work: Yes.

Reproducibility: Yes

Additional Feedback: Overall, this paper studies an important research question regarding the learning behavior of convolutional neural networks. The paper is very well written, seemingly involves a massive amount of wordload, and answers most of the questions clearly with evidence and offer conjectures of the unanswerable questions to guide future research. Despite the high quality, I noticed several drawbacks and suggest the authors to address them. Major issues: -. In the abstract, the paper says the differences "arise not from differences in their internal workings, but from differences in the data that they see", which seems to suggest that whether the model learns texture or shape primarily depends on the data seen, yet in the experiments, the authors demonstrate that, with more carefully designed regularizations (termed as "self-supervised losses" in the paper), the model can be pushed to focus more on the shape. This empirical observation seems to contradict with the main claim in the abstract since I suppose losses are one of the "internal workings" (or what does "internal workings" mean exactly?). I suggest the authors to revise corresponding texts to reflect this more accurately. -. In Section 8, I'm not sure why the authors only study three parts of the AlexNet and two parts of the ResNet, why not report Figure 5 for every single representations one can get from these networks (this study does not seem to introduce new computational loads). I believe it's fine if the authors cannot address this thoroughly and consider this section as an add-on to a good paper, but then I will recommend the authors to stay humble with the conclusions in this section, for example, to rephrase line 62-65 which highlights this section. Also, the authors wrote "suggesting that these models’ classification layers remove shape information", what does "remove" mean exactly? I suppose the last layer has no ability to "remove" or "alter" any preceding representations, it can only prefer one over the other. -. In addition to the dataset mentioned in Section 3, a recent manuscript [1] introduced a new concept of shape-texture dichotomy through the frequency spectrum. I wonder if this main arugments of this paper also follows this understanidng of shape vs. texture, why or why not. Minor issues: "Broader Impact" section should not be numbered. Overall, I think this is a good paper, but I still suggest the authors to address the above issues to further improve the paper. [1] "High-frequency Component Helps Explain the Generalization of Convolutional Neural Networks." Proceedings of the IEEE/CVF Conference on Computer Vision and Pattern Recognition. 2020. ------------------ After reading the rebuttal, I appreciate the authors addressing some of my concerns, my ratings stay unchanged and congrats to overall positive ratings.


Review 4

Summary and Contributions: This paper scrutinizes the origin of texture bias in convolutional neural networks. To do so, it presents some hypotheses for the existence of bias such as network architecture, learning process, or the data. And the answer to each hypothesis is revealed by experiments. This study can be regarded as a good follow up study of former paper(Robert Geirhos et al, 2019), where CNN models are heavily biased towards the texture when trained on ImageNet. As the former finds the 'phenomenon', this study explores some clues that make it happens.

Strengths: As the existence of texture bias in the convolution neural network(CNN) has been known in the machine learning society from a recent study, there is a lot of doubt about why it exists, where it comes from, and how to deal with it. I believe that this is a good empirical study that is needed to understand the bias and answer to them. Also, this study presents how several methods(that are believed to enhance performances on classification tasks) are related to the texture or shape bias of the model. And it can be a good reference for machine learning researchers to guide their model to be more robust and human-like recognition.

Weaknesses: This study provides limited theoretical contributions and is mainly supported by empirical experiments. It would make the arguments more logical if some mathematical proofs are followed.

Correctness: This study has the good design of experiments for finding proofs for each hypothesis. No flaws found.

Clarity: This paper is well written and easy to read. It starts with several questions on texture bias and findings from experiments via hypothesis. Empirical studies have proper figures and tables that make readers understand well.

Relation to Prior Work: As mentioned above, this paper is an in-depth following study of the former where CNN models are biased to texture. And I believe there is no prior work similar to this study.

Reproducibility: Yes

Additional Feedback: ===================== Comments after the reponses of authors: I appreciate authors for all the sincere responses addressing reviewers' comments. However, some of my concerns(e.g. mathematical proofs) still remain the same. Therefore, my overall score would not be changed. ===================== Though this study focuses on the classification task, it can be extended to other tasks where CNN is utilized such as object detection or segmentation. By comparing results among different tasks, we would find whether the origin and prevalence of texture are task-specific or not.

[Author Response · NeurIPS 2020]

We thank the reviewers for their feedback, and their positive appraisal of our work. We respond to each point below.

**R1**: *"I find your experiments extremely interesting with good results and insight, but unfortunately I don't think the*
*current state of the [figure] presentation is publication-quality."* These fixes are easily made: we will use text rather
than WordNet IDs in Fig. 1, increase the font sizes, and add bolding to Tables 2-4 to bring out the key results. We have
updated Fig. 2, reducing the widths of the error bars and grouping results by dataset rather than by model architecture,
and Fig. 3, adding a color gradient to our markers to indicate the additive accumulation of augmentations. We retained
marker shapes for accessibility of colorblind readers and to ensure unambiguous mapping to the legend.

**R1**: *"In Figure 5, I would like to see intermediate (not after the final conv) layers included."* **R3**: *"...why not report*
*Figure 5 for every single representations one can get..."* We initially focused on the output of the convolutional layers
since this is the convolutional representation most commonly used for transfer to other computer vision tasks and
most correlated with IT activity in the primate visual system. We have extended our analysis to consider each layer of
AlexNet. The results recapitulate the ones reported: shape is persistently more decodable through the convolutional
layers than is texture, and its decodability varies less than that of texture, which rises through them. In the FC layers,
shape decodability decreases markedly whereas texture increases.

**R1**: *"Why did you opt for Exemplar as one of your methods in S6?"* Our choice of models was motivated by a desire
for diversity of approaches that perform reasonably well on ImageNet, as well as desire to connect to the existing recent
literature. The Rotation and Exemplar models were recently compared in Kolesnikov et al. 2019. Although there are
differences in implementation, Exemplar, Instance Discrimination, and SimCLR are all constrastive learning methods
and might be expected to behave similarly modulo augmentation. We are unaware of recent work achieving success
with ordinary GANs or VAEs for unsupervised representation learning on ImageNet.

**R2**: *"On the broader impact section: I miss any discussion of the negative impact this work can have or rather that*
*some of the main findings imply."* We have added the following text: "People...often have a mental model of computer
vision models as similar to human vision. Our findings contribute to a body of work showing that this view is actually
far from correct, especially for ImageNet, one of the datasets most commonly used to train and evaluate models.
Divergences between human and machine vision of the kind we study could cause users to make significant errors in
anticipating and reasoning about the behavior of computer vision systems...At the same time, we recognize the possible
negative consequences of blindly constraining models' judgments to agree with people's: human visual judgments
display many forms of bias that should certainly be kept out of computer models." Per **R3**'s pointer, we have removed
the numbering before "Broader Impact."

**R3**: *"what does 'remove' mean exactly?"* In Figure 5, we presented results showing that shape information is down-
weighted by (becomes less decodable through) the fully connected layers of the AlexNet classifier, whereas texture
remains relatively constant. Our statement that "these models' classification layers remove shape information" refers to
the fact that shape decodability is higher at the final convolutional layer (pool3) than in the fully-connected layers, and
for each fully-connected layer, shape decodability is higher for its input than its output. We will add clarifying text.

**R3**: *"In the abstract, the paper says the differences 'arise not from differences in their internal workings, but from*
*differences in the data that they see'...yet in the experiments, the authors demonstrate that, with more carefully designed*
*regularizations...the model can be pushed to focus more on the shape."* Our goal with this statement was to emphasize
the main finding of our study: while factors such as loss function and model architecture do indeed influence the level
of texture bias in a model, they do so to a much lesser degree than do features of the data, e.g. data augmentation.
However, we agree that the statement is worded too strongly. We have added a qualifier: "may not arise primarily from."

*Comparison with other work.* **R1**: *"I'd be interested in a comparison of your architecture findings to those of Ilyas et al."*
Ilyas' main architecture analysis is Fig. 3, which investigates the transferability of adversarial examples from ResNet-50
to other architectures. While differences in the set of models studied unfortunately prevents a complete comparison, we
do observe a qualitative similarity to their results: just as they find ResNet-50 adversarial examples to transfer better to
DenseNet than Inception-v3, we find the shape and texture preferences of ResNet-50 to be more similar to the DenseNet
variants that we study than to Inception-v3. **R3**: *"I wonder if this main arguments of [Wang et al. (2020)] also follows*
*this understanding of shape vs. texture, why or why not."* This is an interesting question. Several studies have found
that CNNs are sensitive to the Fourier statistics of training data and can use high-frequency features imperceptible to
people (also Yin et al. 2019). We see this as related to texture bias, and in our data augmentation experiments, find that
Gaussian blur, which removes high-frequency information, reduces texture bias. At the same time, we do not see shape
and texture as entirely reducible to spatial frequency information (see e.g. Portilla and Simoncelli 2000).

*Textual edits.* Thank you to **R1** for noticing several errors, which we have corrected. *Future follow-up questions.* **R2**
notes that an interesting future follow-up experiment would be to fine-tune the self-supervised models using the same
data augmentations used in the cumulative experiment. We agree – thank you for this suggestion! We also agree with
**R4** that it would be interesting to investigate texture bias in object detection or segmentation models.

[Meta-Review · NeurIPS 2020]

This paper studies an important question and the reviewers found the results convincing enough about the main claim of data augmentation mechanisms contributing to texture bias.